# *Djhsp60* Is Required for Planarian Regeneration and Homeostasis

**DOI:** 10.3390/biom12060808

**Published:** 2022-06-09

**Authors:** Kexue Ma, Rui Li, Gege Song, Fangying Guo, Meng Wu, Qiong Lu, Xinwei Li, Guangwen Chen

**Affiliations:** 1Department of Basic Medicine, Luohe Medical College, Luohe 462002, China; luqiong99999@163.com (Q.L.); txl318@163.com (X.L.); 2College of Life Sciences, Henan Normal University, Xinxiang 453007, China; 18937363112@163.com (R.L.); gfyxinxin@163.com (F.G.); zzfgxj270204@163.com (M.W.); 3Beijing Institute of Genomics, Chinese Academy of Sciences, Beijing 100101, China; 18848976828@163.com

**Keywords:** planarian, regeneration, HSP60, mitochondria, cathepsin L

## Abstract

HSP60, a well-known mitochondrial chaperone, is essential for mitochondrial homeostasis. HSP60 deficiency causes dysfunction of the mitochondria and is lethal to animal survival. Here, we used freshwater planarian as a model system to investigate and uncover the roles of HSP60 in tissue regeneration and homeostasis. HSP60 protein is present in all types of cells in planarians, but it is relatively rich in stem cells and head neural cells. Knockdown of HSP60 by RNAi causes head regression and the loss of regenerating abilities, which is related to decrease in mitotic cells and inhibition of stem cell-related genes. RNAi-HSP60 disrupts the structure of the mitochondria and inhibits the mitochondrial-related genes, which mainly occur in intestinal tissues. RNAi-HSP60 also damages the integrity of intestinal tissues and downregulates intestine-expressed genes. More interestingly, RNAi-HSP60 upregulates the expression of the cathepsin L-like gene, which may be the reason for head regression and necrotic-like cell death. Taking these points together, we propose a model illustrating the relationship between neoblasts and intestinal cells, and also highlight the essential role of the intestinal system in planarian regeneration and tissue homeostasis.

## 1. Introduction

Freshwater planarians are unique animals; they can regenerate a small new animal from a tiny fragment of their bodies. Their extraordinary regeneration capabilities are based on a population of pluripotent stem cells, known as neoblasts [1,2]. Neoblasts are small undifferentiated cells located in parenchymal tissues [2,3]. After amputation, neoblasts proliferate to produce multiple cell types for regenerating the missing tissues [4]. Planarian regeneration is a complicated cellular process, and it involves not only neoblast proliferation, migration, and differentiation, but also apoptosis and autophagy occurring in the old tissues [5,6,7]. It should be noted that all cellular activity needs ATP produced by the mitochondria to provide energy. Therefore, maintaining mitochondrial homeostasis is very important for planarian regeneration and tissue homeostasis.

Mitochondria are important organelles in eukaryotic cells. The three main functions of mitochondria are to generate ATP, buffer cytosolic calcium, and produce reactive oxygen species (ROS) [8]. Although the mitochondria have their own genome, ATP production mainly relies on the proteins encoded by the nuclear genome [9]. These proteins need chaperones to help them fold and unfold before and after being imported into the mitochondria [10]. In addition, the folding of mitochondrial proteins is also affected by ROS [11]. ROS is a by-product of oxidative phosphorylation (OXPHOS), which can trigger the unfolded protein response in mitochondria (MT-UPR) and further disturb mitochondrial function [12]. To help proteins fold into their native conformation and avoid unwanted protein–protein interactions or aggregation in mitochondria, several heat shock proteins (HSPs) play important roles in mitochondrial proteostasis [13,14].

HSP60, a well-known mitochondrial chaperone, has been shown to promote protein folding in the mitochondria. The HSP60 chaperone is located in the matrix and consists of both HSP60 and HSP10 subunits to form a barrel-shaped complex that has been suggested to facilitate the folding of relatively small, soluble monomeric proteins [15]. Knockout experiments have demonstrated that HSP60 is essential for survival in *Drosophila* [16]. In humans, a deficiency in HSP60 has been associated with neurodegenerative diseases [17,18,19]. HSP60 is more abundant in mouse embryonic stem cells (ESCs) than in mouse embryonic fibroblasts, and the depletion of HSP60 inhibits mouse ESC proliferation and self-renewal [20]. HSP60 deficiency in adult mice causes mitochondrial dysfunction accompanied by impaired proliferation of the intestinal epithelium, and loss of stemness [21]. HSP60 involvement in maintaining stem cell self-renewal may be related to mitochondrial function. HSP60 can also bind with CCAR2 (cell cycle and apoptosis regulator protein 2), which is essential for maintaining mitochondrial membrane potential [22]. HSP60 deletion in adult mice hearts altered mitochondrial complex activity, mitochondrial membrane potential, and ROS production [23]. Therefore, HSP60 plays a critical role in maintaining mitochondrial function.

Although many studies have revealed that HSP60 is necessary for tissue regeneration [24,25], the underlying mechanism is not well elucidated. After the amputation of tissues or organs, cells adjacent to the wound site are exposed to a variety of stresses, such as the generation of ROS [26]. Heat shock proteins expressed in wound sites are supposed to protect cells from damage [27,28]. Many studies have shown that during planarian regeneration, *hsp* family genes were highly expressed in the blastema [29,30]. Mortalin, a member of the HSP70 family located in the mitochondria, shows a stem cell-like expression pattern in planarians [30]. Knockdown of mortalin impairs planarians’ regeneration ability, suggesting that it is crucial for planarian stem cell viability [30]. Although previous work has revealed that HSP60 transcripts were constitutively expressed in planarian stem cells [29], its function was not well investigated. In our work, we present the following findings: (1) The transcripts of HSP60 are mainly distributed in parenchymal tissue (where planarian stem cells are located) with a stem cell feature, but HSP60 protein is ubiquitously expressed in all types of cells; (2) RNAi-HSP60 impairs planarian regeneration ability, and decreases the expression of stem cell-related genes; (3) RNAi-HSP60 inhibits the expression of mitochondrial-related genes, disrupts mitochondrial function, and destroys the structure and function of the intestinal system; and (4) RNAi-HSP60 activates cathepsin L-mediated necrotic cell death. Our work highlights the essential role of the intestinal system in planarian regeneration and homeostasis.

## 2. Materials and Methods

### 2.1. Animals

Planarians *D. japonica* were collected from Yuquan (Hebi City, China) and were asexually reproduced in our laboratory. They were reared at 20 °C and starved for at least 7 days before the experiments were performed. Animals of 4–6 mm length were used for RNAi, and animals of 2–4 mm length were used for in situ hybridization.

### 2.2. Identification of Djhsp60 and RNAi Experiments

The full-length cDNA sequence of *Djhsp60* was obtained from our transcriptome data of *D. japonica*, which has been deposited in Genbank (accession number: MH509193). Double-stranded RNA (dsRNA) synthesis and RNAi experiments were performed as previously described [31]. Briefly, 10 μg dsRNA mixed with 20 μL beef liver puree was used for feeding 20–25 animals. Animals were fed 5 times over the course of 17 days. Animals with mild head-defect and moderate head-defect were collected for further studies. Images of live animals were captured with a Leica DFC300FX camera and processed in Adobe Photoshop. DsRNA for GFP was used as negative control, dsRNA for *Dj-β-catenin-*1 was used as positive control. To complete this work, RNAi experiments were repeated over 8 times. The primers used in this work were listed in Appendix A.

### 2.3. QRT-PCR

Total RNA was extracted using Trizol reagent (Invitrogen, Cat. 15596026, Waltham, MA, USA) and digested with RNase-free DNase I (TaKaRa, Dalian, China), and 2 μg of total RNA was used for the first-strand cDNA synthesis (TaKaRaPrimeScript™ II 1st strand cDNA synthesis kit, Code No. 6210A) based on the manufacturer’s protocol. QRT-PCR was performed with ABI PRISM 7500 Sequence Detection System (Applied Biosystems, Waltham, MA, USA) using the TB green™ premix Ex Taq™Ⅱ kit (Takara, Dalian, China). The total volume of 20 μL reaction mixture consisted of 10 μL 2 × TB Green Premix Ex Taq Ⅱ (ROX plus), 0.8 μL gene-specific forward and reverse primers (10 μM each), 1 μL cDNA template, and 7.4 μL RNase-free water. Reaction conditions were 95 °C for 5 min followed by 40 cycles of 10 s at 95 °C, 30 s at 60 °C. Planarian *elongation factor 2* (*Djef*2) was utilized as the reference gene in all of the experiments [32]. Expression ratios were determined with the 2^−ΔΔ^^CT^ method, which was described by Livak and Schmittgen [33]. The primers used in this work were listed in Appendix A.

### 2.4. In Situ Hybridization

The following DIG- (Roche, Basel, Switzerland) or FITC- (Roche, Basel, Switzerland) labeled riboprobes were synthesized using an in vitro transcription kit (SP6/T7, Roche). Colorimetric whole-mount in situ hybridization (WISH) and fluorescent in situ hybridization (FISH) were performed as previously described with a minor revision [32]. In brief, planarians were killed and immediately fixed with 4% formaldehyde in PBST. After bleaching in 6% H_2_O_2_, the samples were treated with proteinase K and were re-fixed with 4% formaldehyde. After the above treatment, the samples were incubated in 400 ng/mL Riboprobe mix > 16 h at 56 °C. After stringency washing and blocking were performed, the samples were incubated in anti-digoxigenin alkaline phosphatase-conjugated antibody (1:2000, Roche, 11093274910) overnight at 4 °C. For WISH, hybridization signals were detected using BCIP/NBT standard substrates. For FISH or double-FISH analyses, samples were incubated overnight in anti–DIG-POD (1:2000 dilution; Roche, Catalog No. 11207733910) and anti-fluorescein-POD (1:2000 dilution; Roche, Catalog No. 11426346910), developed using Rhodamine-tyramide and FITC-tyramide solution. DAPI (5 μg/mL) was used as the final step to reveal the CNS structure. Images were captured on a Leica DFC300FX camera and processed in Adobe Photoshop. Fluorescent images were collected on stereo fluorescence microscope (Axio Zoom. V16, Zeiss) and fluorescence microscopes (Image.A2, Zeiss).

### 2.5. Paraffin Section, HE Staining and Immunochemistry

Animals were killed in 2% HCl and immediately fixed in 4% formaldehyde PBS at 4 °C for 6 h. Samples were dehydrated through a graded series of ethanol and embedded in paraffin. Tissue sections of 6-μm thickness were mounted on slides. Slides were deparaffinized and rehydrated in graded alcohols and washed in PBST. For histological analysis, slides were stained with Hematoxylin and Eosin (H&E), and images were scanned with Caseviewer 2.0 (Pannoramic 250/MIDI). For immunochemical analysis, slides were incubated in 10% block reagent (Roche, Catalog No. 11921673001). After removing the block reagent, slides were incubated in HSP60 primary antibody (ADI-SPA828-D, Enzo Life Sciences, 1:30 dilution) at 37 °C for 1 h. After washing in PBST 3 times for 10 min, sections were incubated in rabbit anti-goat Alexa Fluor 594 secondary antibody (1:200 dilution) at 37 °C for 30 min, followed by incubation in DAPI (5 μg/mL) for 10 min. A negative control with PBST instead of HSP60 primary antibody was used, and the specificity of primary antibody was validated by western blot (Appendix A). Fluorescent images were collected on fluorescence microscopes (Image.A2, Zeiss) and Leica TCS SP2 confocal microscope and processed in Adobe Photoshop.

### 2.6. Immunostaining on Dissociated Cells

Cell dissociation was performed as previously described [34]. Planarians were cut into three to four fragments on ice with a scalpel. The resultant animal fragments were soaked in PBS. The fragments were cut into smaller pieces and treated with 0.25% (*w*/*v*) trypsin for several minutes at 20 °C. The samples were completely dissociated into single cells by gentle pipetting. The cell mixture was then fixed in 4% formaldehyde PBS for 30 min and dropped on slides to dry at 37 °C for 2 h. The slides were washed in PBST 2 times for 5 min, and incubated in 10% block reagent for 20 min. Immunostaining for HSP60 was conducted as described above. Fluorescent images were collected on fluorescence microscopes (Image.A2, Zeiss).

### 2.7. Whole-Mount Immunostaining

Whole-mount immunostaining was performed as described [35]. Anti-phosphorylated histone-3 (H3P) primary antibody (Millipore 06-870, Burlington, MA, USA, 1:200 diluted in 1 × PBS with 0.3% triton X-100) was used to detect the mitotic cells, goat anti-rabbit Alexa Fluor 594 was used as secondary antibody in this experiment. Fluorescence signals were detected with a Stereo fluorescence microscope (Axio Zoom. V16, Zeiss). To determine the mitotic activity of the neoblasts, the total number of H3P positive cells was determined using ImageJ by measuring the surface of the animals before sampling.

### 2.8. Whole-Mount TUNEL

Animals were fixed and stained for TUNEL as previously described [36] using the ApopTag Red In Situ Apoptosis Detection Kit (CHEMICON, S7165) with some modifications. After bleaching, samples were incubated for 6 h in terminal transferase enzyme at 37 °C, and again overnight at 4 °C with anti-dioxigenin-rhodamine. Images were scanned, processed and quantified as described for FISH images. To avoid technical variance and obtain a reliable quantification of TUNEL^+^ cells, at least 5 animals were used for statistics.

### 2.9. RNA Sequencing and Gene Expression Analysis 

Total RNA was extracted from control and RNAi animals using the mirVana miRNA Isolation Kit (Ambion-1561) protocol. The libraries were constructed using TruSeq Stranded mRNA LTSample Prep Kit (Illumina, San Diego, CA, USA). They were sequenced by Illumina Nova-seq with paired-end 150 bp read length. The transcriptome sequencing and analysis were conducted by OE Biotech Co., Ltd. (Shanghai, China). Briefly, all Illumina sequencing was deposited in the Sequence Reads Archive as study PRJNA754705. The function of the unigenes was annotated by NCBI nonredundant (NR), SwissProt, and Clusters of orthologous groups for eukaryotic complete genomes (KOG) databases using Blastxwith a threshold E-value of 10^−5^. Based on the SwissProt annotation, Gene ontology (GO) classification was performed by the mapping relation between SwissProt and GO term. The unigenes were mapped to the Kyoto Encyclopedia of Genes and Genomes (KEGG) database to annotate their potential metabolic pathways. FPKM (Fragments Per Kilobase Million) and read counts value of each unigene was calculated using bowtie 2 [37] and eXpress [38]. DEGs were identified using the DESeq functions estimate Size Factors and nbinomTest [39]. *p* value < 0.05 and foldChange > 2 or foldChange < 0.5 was set as the threshold for significantly differential expression. GO enrichment and KEGG pathway enrichment analysis of DEGs were respectively performed using R based on the hypergeometric distribution. Animals with head-defect were collected at 25 days after RNAi for RNAseq analysis.

### 2.10. Assessment of Mitochondrial Respiration

Mitochondrial respiration will produce reactive oxygen species (ROS), which can be detected by H_2_DCFDA [26]. Therefore, only normal mitochondria can produce ROS, whereas damaged mitochondria do not. In this work, we used H_2_DCFDA staining as an index to assess mitochondrial function indirectly. Planarians were incubated in culture water containing H_2_DCFDA (25 μM, 1 mL) for 30 min, then they were immediately placed on iced black hard paper (ventral side facing up) for scanning by a Stereo fluorescence microscope (Axio Zoom. V16, Zeiss). Then, 5 animals were randomly selected from each group for detection. Under our experimental conditions, the branches of intestinal system in the control animals could be clearly revealed by H_2_DCFDA staining, and the good images of living organisms were easily obtained.

### 2.11. Transmission Electron Microscopy (TEM)

TEM was performed as previously described [32]. In brief, the planarians were immediately fixed in 2.5% *v*/*v* glutaraldehyde at 4 °C for 24 h. After the samples were washed with PBS, secondary fixation was performed with 1% *w*/*v* OsO4. The samples were dehydrated with a graded alcohol series to 100% and with acetone. Afterwards, the samples were embedded in Epon 812 resin mixture. Tissue cross-sections were cut at a thickness of 70 nm with an ultramicrotome and stained with uranyl acetate and lead citrate. Ultrathin sections were photographed with a TEM (Hi-7700, Hitachi, Japan). Three animals were randomly selected from each group (the control and RNAi, respectively) for TEM.

### 2.12. Statistical Analysis

Statistical analysis was performed using SPASS 14.0. Data were subjected to one-way ANOVA. Differences were considered significant at *p* <0.05 level and extremely significant at *p* <0.01 level.

## 3. Results

### 3.1. Expression Patterns and Protein Localization of HSP60 in the Planarian Dugesia japonica

To elucidate the function of HSP60 during planarian regeneration, we first isolated the full-length cDNA of HSP60 from our transcriptome data of *D. japonica*. It encodes a polypeptide of 575 amino acids with a predicted molecular mass of 61.68 kDa (Designated as DjHSP60, Genbank accession No. MH509193). DjHSP60 has six conserved GGM repeats at its C-terminal end (Appendix A), which is typical of mitochondrial HSP60s. The transcripts of *Djhsp60* were highly expressed in parenchymal tissue (the regions that neoblasts are thought to be present) [1,2], and weakly expressed in differentiated tissues, such as the head, pharynx, and intestinal tissues (Figure 1A). The expression patterns of *Djhsp60* in intact animals were very similar to those of *DjwiA* (the homologue of *Smedwi-1*, a widely used stem cell marker in planarian) [1]. After amputation, the expression of *Djhsp60* increased significantly (Figure 1B). The strong positive signals first appeared at the cutting site on Day 1 of regeneration, then focused on the center of a tail fragment on Day 3 of regeneration, subsequently surrounding the newly regenerated pharynx on Days 5–7 of regeneration and then decreased in the newly regenerated head and pharynx (Figure 1B). This expression pattern supports the idea that the transcripts of *hsp60* are upregulated with cell proliferation but downregulated with cell differentiation. To further examine the expression patterns of *Djhsp60*, we used a double-FISH to test whether *Djhsp60* was co-expressed with neoblast markers. Our results showed that most *Djhsp60*-positive signals were co-expressed with those of *DjwiA* (Figure 1C). In addition, we also observed that *Djhsp60*/*DjH2B* (pan-neoblast marker) co-expressed cells nearly reached 100% (Figure 1C). However, *Djhsp60*-positive signals were also co-expressed with those of the epidermal progenitor marker *NB21.11e* (Figure 1C), which suggested that the expression of *Djhsp60* was not neoblast-specific. To further validate this idea, we blasted the expression pattern of *Smedhsp60* using the single cell data (https://digiworm.wi.mit.edu/, accessed on 5 April 2022) [40]. The results showed that *Smedhsp60* was expressed in parenchymal tissues, intestine, pharynx, epidermis, muscle, neural tissues, and smedwi-1^+^ cells (Appendix A). We next examined the protein localization of DjHSP60 through fluorescent immunostaining of prepared paraffin sections and dissociated cells. The results showed that DjHSP60 was ubiquitously distributed in all types of tissue (Figure 2A–G) but was highly expressed in head neural tissues (Appendix A) and parenchymal tissues (where the neoblasts are distributed). Fluorescent immunostaining also showed that *DjHSP60*-positive signals were localized in cytoplasm in dissociated cells (Figure 2H), which is a characteristic of mitochondrial protein. In addition, *DjHSP60*-positive cells were small in size (about 6–10 μm) with scanty cytoplasm, which is a characteristic of neoblasts [2]. These results confirmed the idea that *hsp60* is not a neoblast-specific marker in planarians.

### 3.2. RNAi-Djhsp60 Affects Planarian Regeneration and Tissue Homeostasis

To investigate the function of *Djhsp60* in planarians, we used RNA interference (RNAi). Initially, animals were cut after 14 days of feeding on double-stranded RNA (dsRNA), and most of them could regenerate normal heads. It seems that RNAi does not affect planarian regeneration (Appendix A). However, the newly regenerated animals showed loss of auricles, and the heads became round after 20 days of regeneration (Appendix A). In order to collect phenotype-defect animals, we revised the RNAi plan (Figure 3A). We found that some planarians began to show regression of one side of the head, then the whole head in the following days after 20 days of RNAi (Figure 3A). The head-defective animals moved slowly and showed insensitivity to light (Appendix A). DAPI staining showed severe degeneration in the structure of CNS (Figure 3A). We observed that phenotype defects after RNAi progressed through temporal stages, ending in lethality. For example, 20 days after RNAi, the number of phenotype defects was over 20% (5/23); 25 days after RNAi, the number of phenotype defects was over 60% (15/23); and 30 days after RNAi, the number of phenotype defects was over 80% (20/23, Figure 3B). More importantly, most of the animals with defective phenotype were incapable of regeneration. Some only regenerated small blastema (Figure 3C). To test the efficiency of our RNAi experiment, we detected the endogenous mRNA expression levels of *Djhsp60*. The qPCR results showed that the mRNA expression levels of *Djhsp60* after RNAi were reduced to approximately 40% of the control level (Figure 3D). Consistent with the qPCR results, the WISH results showed that most of the *Djhsp60* transcripts were removed after RNAi (Figure 3E). These results suggest that *Djhsp60* is required for planarian regeneration and tissue homeostasis. 

### 3.3. Effects of RNAi-Djhsp60 on Stem Cell Viability

Head regression and regeneration deficiency are likely due to the loss of neoblasts. Since *Djhsp60*-RNAi animals resemble irradiated animals and *Smedwi-2*-RNAi animals [1], *Djhsp60* is likely needed for neoblast function. To test this hypothesis, we detected the expression of neoblast markers after RNAi in intact and regenerating animals. Our results showed that the *DjwiA*-positive cells disappeared from the whole body and only small positive cells remained in the tail after feeding with dsRNA (Figure 4A), and the expression levels of *DjwiA* were reduced to 10% of the control levels (Figure 4A). We further detected the expression levels of *DjMCM2* (another widely used stem cell marker in planarians) and *DjH2B* (pan-stem cell marker in planarian) in regenerating animals [41,42]. Their transcripts were barely detected in the trunk and tail fragments (Figure 4B), and their expression levels were reduced to 10% to 15% of the control levels (Figure 4B). To further verify that RNAi-*Djhsp60* disturbed neoblast proliferation, we analyzed the mitotic marker phospho-histone-H3 (H3P) in RNAi animals. Immunostaining with anti-H3P demonstrated that the number of mitotic cells was dramatically reduced in intact RNAi worms (52 ± 6/mm^2^) in comparison with control worms (201 ± 33/mm^2^) (Figure 4C). After amputation, two mitotic peaks (6 h post-amputation and 48 h post-amputation) could not be observed compared with the controls (Figure 4D). These results suggest that *Djhsp60* is necessary for neoblast self-renewal and proliferation.

### 3.4. RNAi-Djhsp60 Caused Necrotic Cell Death

Head regression after RNAi was likely due to cell death. To test this hypothesis, we used whole-mount TUNEL staining to detect the amount of cell death. We observed a significant increase in TUNEL-positive cells in *Djhsp60*-RNAi animals (Figure 5A). The amount of cell death was 9 ± 3/0.1 mm^2^ in control animals but 186 ± 26/0.1 mm^2^ in RNAi animals (Figure 5B). It should be noted that TUNEL staining only detects DNA strand breakage, which occurs in any type of cell death [43]. To determine which type of cell death occurred in RNAi animals, we used TEM to observe the morphological features of dead cells. Our results showed that the features of cell death in RNAi animals was very similar to that of necrotic cell death [44], including the loss of the nuclear envelope, the condensation of chromatin, the appearance of nuclear fragments, and the formation of intracellular vacuoles (Figure 5C). These features of cell death were very different from those of autophagic cell death and apoptotic cell death [32,45].We further used HE staining to reveal the cell morphology. As tissue degradation after RNAi first appeared on two sides of the head, we deduced that cell death may occur in this region. HE staining verified this hypothesis. We observed that many dead cells with the features of necrotic-like cell death are located on the two sides of the head of RNAi samples (Figure 5D). By contrast, no typical dead cells were observed in the head region of the control samples (Appendix A). In summary, these results confirmed that RNAi-*Djhsp60* caused necrotic-like cell death in planarians.

### 3.5. RNAi-Djhsp60 Globally Inhibits Gene Expression

To explore why RNAi-*Djhsp60* caused the loss of regeneration abilities and tissue homeostasis, RNA-seq was used to analyze the differentially expressed genes after RNAi. The results showed that 2858 of 37,288 transcripts (approximately 7.7% coverage) were downregulated in RNAi animals compared with the controls, whereas only 34 transcripts were upregulated (Figure 6A). We searched for the top 20 KEGG enrichment terms of the downregulated transcripts and found that a large number of transcripts were related to mitochondrial function and protein synthesis (Figure 6B, Appendix A). The KEGG analysis revealed that RNAi-*Djhsp60* may disrupt mitochondrial function and lead to a shortage of ATP and proteins needed for neoblast proliferation (Figure 6B). To verify our hypothesis, we blasted the downregulated transcripts, and found that neoblast-related transcripts were downregulated (Appendix A). We further used qRT-PCR to detect the cell cycle-related genes, and found that the genes related to both the G1→S phase (Cyclin D and E2F) and G2→M phase (Cyclin B and CDK1) transitions were significantly reduced (Figure 6C). Therefore, we speculated that RNAi-*Djhsp60* affected neoblast proliferation due to mitochondrial dysfunction.

### 3.6. RNAi-Djhsp60 Downregulates the Genes Related to the Glycolysis and Pentose Phosphate Pathway

Although our results suggested that RNAi-*Djhsp60* impaired neoblast proliferation, the underlying mechanism remained unknown. We speculated that RNAi-*Djhsp60* may damage the energy metabolic pathways for neoblast division. ATP production in stem cells is more dependent on glycolysis, and nucleotide synthesis relies on the pentose phosphate pathway (PPP). We blasted the downregulated transcripts, and found that the transcripts related to glycolysis and PPP were downregulated (Appendix A). Hexokinase (HK), pyruvate kinase (PK), and phosphofructokinase 1 (PFK1) are the rate-limiting glycolysis enzymes, and their expression levels in RNAi animals were reduced by 0.71-fold, 0.32-fold, and 0.43-fold, respectively (Figure 7). Glucose-6-phosphate dehydrogenase (G6PD), ribose-5-phosphate isomerase (RPI), and transketolase (TKT) are key enzymes of PPP, and their expression levels in RNAi animals were reduced by 0.91-fold, 0.34-fold, and 0.46-fold, respectively (Figure 7). Glucose intake is also important for energy metabolism in neoblasts. Our results showed that the mRNA expression levels of glucose transporter 1 (GLUT1) and glucose transporter 2 (GLUT2) were reduced by 0.29-fold and 0.18-fold, respectively (Figure 7). These results suggest that RNAi-*Djhsp60* may impair the energy metabolism of neoblasts.

### 3.7. RNAi-Djhsp60 Severely Affects Mitochondrial Function

KEGG pathway analysis revealed that a large number of transcripts related to mitochondrial function were downregulated (Figure 6B). We blasted the downregulated transcripts and found that 110 of 2512 transcripts (>−1.5-fold) were mitochondrial transcripts. Of these, 41 transcripts belonged to mitochondrial electron transport chain (ETC) complexes (Appendix A). To validate them, we selected several transcripts to detect their expression after RNAi (Figure 8A). Citrate synthase (CS), isocitrate dehydrogenase (IDH), and 2-oxoglutarate dehydrogenase (OGDH) play an important role in the tricarboxylic (TCA) cycle of the mitochondria of all organisms and are widely used as indicators to assess mitochondria function [46,47]. Our results showed that their transcripts after RNAi were reduced by 0.33-fold, 0.39-fold, and 0.32-fold, respectively. Acyl-CoA dehydrogenase (ACAD), an important mitochondrial enzyme, participates in consecutive cycles of β-oxidation to generate acetyl-CoA for generating energy [48]. The expression levels of ACAD transcripts after RNAi were reduced by 0.57-fold compared with the control level. The transcripts of other mitochondrial enzymes (including aspartate aminotransferase, ETC complexes) were reduced significantly after RNAi compared with the control level. Mortalin, another mitochondrial chaperone (mt-Hsp70), functions in maintaining mitochondrial homeostasis [49]. Its transcripts after RNAi were reduced by 0.63-fold compared with the control. Bcl2, a well-known anti-apoptotic protein localized in the mitochondria, promotes cell survival and proliferation in various cell types [50,51]. Our results showed that the expression levels of *Bcl2* after RNAi were reduced by 0.29-fold compared with the control level. Mitochondrion-related genes were greatly downregulated, suggesting that RNAi-*Djhsp60* severely impaired mitochondrial function. To confirm this hypothesis, we used H_2_DCFDA to assess the mitochondrial respiration indirectly. Mitochondrial respiration produces ROS, which can be detected by H_2_DCFDA [26]. Therefore, only living cells can be revealed by H_2_DCFDA staining. Our results showed that the intestinal system was clearly revealed by H_2_DCFDA staining in the control animals, whereas the signals of H_2_DCFDA disappeared gradually from the intestinal system in RNAi animals with the progression of RNAi (Figure 8B). The H_2_DCFDA staining experiment suggested that mitochondrial respiration was impaired and most of the intestinal cells were dead in RNAi animals. The integrity of the mitochondrial inner and outer membrane are necessary for mitochondrial respiration. RNAi-*Djhsp60* may damage the mitochondrial structure. Consistent with this hypothesis, TEM revealed that a large number of damaged mitochondria had accumulated, together with the features of the broken mitochondrial cristae and the loss of two membrane layers (Figure 8C). These results suggest that RNAi-*Djhsp60* impaired the structure of the mitochondria and led to the loss of mitochondrial function.

### 3.8. RNAi-Djhsp60 Severely Damages the Structure and Function of the Intestinal System

The intestinal system is an important “nutrient organ” in planarians, which can provide energy or nutrients for all types of cells [52]. Figure 8B revealed the loss of mitochondrial respiration in the intestinal cells, suggesting that RNAi-*Djhsp60* affected the intestinal function. Recently, Forsthoefel et al. identified 1844 intestine-enriched transcripts [53], which contained 78 mitochondria-related transcripts, suggesting that mitochondria-related transcripts were rich in intestinal cells. We further compared the 1844 intestine-enriched transcripts with 2858 downregulated transcripts after RNAi-*Djhsp60*, and found that over 30% (564/1844) of the intestine-enriched transcripts were downregulated (Appendix A, derived from reference [53]). These data further support the hypothesis that RNAi-*Djhsp60* affects the function of intestinal cells. We detected the expression levels of several intestinal transcripts (Figure 9A) and found that the transcripts of gastric triacylglycerol lipase (Lipf), aspartyl aminopeptidase (Dnpep), solute carrier family 5 (Slc5), and solute carrier family 10 (Slc10) were significantly reduced (** *p <* 0.01). Other transcripts, namely chymotrypsin-like elastase (Cela), carboxypeptidase A2 (Cpa2), fructose-1,6-bisphosphatase (Fbp), were also downregulated (* *p* < 0.05). The WISH results further confirmed the downregulation of gastric triacylglycerol lipase and chymotrypsin-like elastase after RNAi (Figure 9B). In addition, our RNA-seq data revealed the downregulation of antioxidant enzymes, such as superoxide dismutase (SOD), glutathione S-transferase (GST), and glutathione peroxidase (GPX). Our FISH results showed that *Sod*, *Gst**,* and *Gpx* were mainly expressed in the intestinal system (Figure 9C), and the downregulation of their transcripts after RNAi was also validated by qRT-PCR (Figure 9D). Recently, Bijnens et al. [54] investigated ROS production and the expression of antioxidant genes in planarians, and their results confirmed the reliability of our findings. The downregulation of antioxidant genes and the loss of ROS in the intestinal system both indicated that the intestinal system was severely affected by RNAi. To validate this deduction, we observed the morphological features of planarian intestinal tissues. HE staining sections showed that the intestinal branches were arranged regularly, and intestinal tissues contained a small number of dark eosinophilic vesicles in the control samples (Figure 9E and Appendix A). However, the integrity of the intestinal structure was destroyed, and the intestinal tissues were degraded into numerous dark eosinophilic vesicles in RNAi samples (Figure 9E and Appendix A). We further detected the expression patterns of DjHSP60 protein after RNAi, and found that the distribution of DjHSP60 protein in RNAi sections was very different from that in control sections (Figure 10A,B, boxed parts). In control sections, it was very difficult to differentiate parenchymal tissues from intestinal tissues (Figure 10A). Conversely, this was very easy in RNAi sections (Figure 10B). When we compared the dark regions of the RNAi sections (Figure 10C) with the HE staining sections (Figure 10D), we found degraded intestinal tissues. Figure 10 indicated that the HSP60 protein levels in intestinal tissues reduced much more quickly than in parenchymal tissues after RNAi, so the parenchymal tissues became evident. In other words, intestinal tissues are more vulnerable to *hsp60* dsRNA exposure than parenchymal tissues. In summary, these data suggested that RNAi-*hsp60* damaged the structure and function of intestinal cells.

### 3.9. RNAi-Djhsp60 Activates Cathepsin-Mediated Cell Death

Very interestingly, of the 34 transcripts upregulated after RNAi (Appendix A), only cathepsin L (CatL) is associated with cell death and tissue degradation [55,56]. The qRT-PCR results showed that the transcripts of *DjCatL* were increased by 3.7-fold after RNAi compared with the control (Figure 11A). CatL is mainly expressed in the intestinal tissue and can be secreted into the extracellular matrix. After RNAi, strong positive signals of *DjCatL* were detected in intestinal branches between the head and pharynx (Figure 11B). As tissue regression begins at the head, we suppose that *DjCatL* may play an important role in this process. In addition, we detected several apoptosis-related transcripts (Figure 11A), and their expression levels were downregulated. Although we cannot exclude the possibility that caspase-mediated cell death occurred in RNAi animals, our results indicated that cathepsin L may be an important death-executer in RNAi-*Djhsp60* animals (Figure 11B).

## 4. Discussion

HSP60, a well-known mitochondrial chaperone, plays an important role in cell survival and cell death. High levels of HSP60 expression are associated with cell proliferation, especially in various types of cancers [57], such as liver cancer, prostate cancer, and colorectal cancer. Inhibition of HSP60 expression can block cell proliferation and lead to cell death [58,59]. Therefore, HSP60 was regarded as a potential target for cancer treatment [57,59]. However, most studies have been conducted in vitro or have used tissues or organs for evaluation. Studying the deficiency in HSP60 at the whole-animal level is more important for the development of novel anti-cancer therapy. In this work, we used planarians as a whole-animal model system to determine HSP60 expression patterns and investigated the effects of RNAi-*Djhsp60* on planarian regeneration and tissue homeostasis. Our findings provide new insights into HSP60 in tissue regeneration and homeostasis, and also contribute to evaluating the choice of HSP60 as a therapeutic target.

### 4.1. HSP60 Is Ubiquitously Expressed in All Types of Cells in Planarians

Previous studies revealed that the transcripts of *Dj**hsp60* were distributed in parenchymal tissues with the features of planarian stem cells [29,60]. We were very interested in whether the expression of HSP60 is neoblast-specific. Our results revealed that *Djhsp60* was not only expressed in parenchymal tissues but also in intestinal tissues, and *Djhsp60* was co-expressed with both stem cell markers and epidermal progenitor markers. In addition, fluorescent immunostaining revealed that HSP60 protein was present in all types of cells in planarians. Very interestingly, HSP60 protein was highly enriched in planarian head neural tissues and parenchymal tissues. As a mitochondrial protein, HSP60 can be present in all eukaryotic cells. Therefore, the expression of HSP60 in planarians is not specific to a particular cell type. Although our findings and other reports have revealed that the upregulation of HSP60 is related to cell proliferation [61,62], the side effects of choosing HSP60 as a therapeutic target should be considered.

### 4.2. RNAi-Djhsp60 May Impair the Energy Metabolism of Planarian Stem Cells

During the past decades, numerous studies into planarian regeneration have focused on the neoblasts [1,2,3,4,5]. It should be noted that ATP production and biosynthesis are needed for all cellular activities, such as cell proliferation and differentiation. In this work, the transcripts of *Djhsp60* were co-expressed with planarian neoblast markers, and HSP60 protein was rich in parenchymal tissues (where the neoblasts are distributed). After 20 days of dsRNA exposure, planarians lost their regeneration abilities. Consistent with this phenomenon, the expression of neoblast markers, and the number of mitoses, were significantly reduced after RNAi. Therefore, *Djhsp60* is necessary for neoblast proliferation. As a mitochondrial chaperone, the main function of HSP60 is to maintain mitochondrial homeostasis. Interestingly, recently studies have highlighted the essential role of mitochondria in planarian regeneration [63]. We also observed that neoblast-like cells contained mitochondria (Appendix A). Furthermore, we observed that the structure and function of mitochondria were damaged in RNAi animals. In addition, both our RNA-seq data and qPCR results revealed that the transcripts related to TCA, glycolysis, and the PPP pathway were downregulated (Appendix A). Therefore, we deduced that RNAi-*Djhsp60* may impair the energy metabolic system in neoblasts. 

### 4.3. Intestinal Cells Play an Essential Role in Planarian Regeneration and Tissue Homeostasis

During the past decades, numerous studies have highlighted the role of neoblasts in planarian regeneration and tissue homeostasis. However, understanding planarian regeneration based on the aspects of energy metabolism and nutrient uptake has been largely ignored. The intestinal system is an important organ for planarians, responsible for nutrient uptake, storage, and turnover. In recent years, several studies have revealed that the disturbance of intestinal function by knockdown of several intestine-enriched transcripts caused reduced blastema formation and/or decreased neoblast proliferation [4,64,65]. However, the inherent relationship between neoblasts and intestinal cells has not been well elucidated. In this work, we demonstrated that RNAi-*Djhsp60* destroyed the structure and function of mitochondria, which mainly occurred in intestinal tissues. Furthermore, we found that the intestinal tissues degenerated into numerous vesicles, and over 30% of the intestine-enriched transcripts were downregulated. Obviously, RNAi-*Djhsp60* damaged the structure and function of intestinal tissues. Under our experimental conditions, we found that intestinal tissues were more susceptible to the loss of HSP60 than parenchymal tissues, because intestinal cells contain numerous mitochondria [66] and rely on OXPHOS to produce ATP. The direct evidence is that HSP60 protein disappears from intestinal tissues after RNAi (Figure 10B,C). According to our findings, it is inferred that intestinal cells cannot provide nutrients (such as glucose, amino acid, lipids, etc.) for neoblasts and other cell types after the loss of function induced by RNAi. Without the nutrients, ATP cannot be produced by glycolysis in neoblasts, which is needed for gene transcription and protein synthesis before neoblasts enter the cell cycle. Therefore, the relationship between neoblasts and intestinal cells is co-dependent. Neoblasts are the basis for intestinal regeneration and tissue-renewal; conversely, intestinal cells provide nutrients for neoblasts to generate ATP to support proliferation, migration, and differentiation (Figure 12A). This idea is supported by the tissue structure. Unlike other animals, the intestinal tissues in planarians are distributed throughout the whole body from head to tail, and closely adjacent to parenchymal tissues (where the neoblasts are located). We observed that neoblast-like cells were surrounded by intestinal cells (Appendix A). In addition, numerous studies have revealed that neoblasts reside near intestinal branches [3,4,67,68], suggesting that the intestine might play a niche-like role in modulating neoblast dynamics [2]. In summary, the dysfunction of intestinal cells induced by RNAi-*Djhsp60* affects the energy metabolism of neoblasts, or alters the niche of neoblasts. This idea needs more detailed investigation.

### 4.4. Disruption of the Mitochondrial—Lysosome Axis and Activating Cathepsin L-Mediated Cell Death after RNAi May Be the Reason for Cell Death

The disturbance of mitochondrial function can induce multiple types of cell death through different pathways [69], and the disruption of the mitochondrial–lysosomal axis, related to a progressive decline in cell function and integrity, has been discussed in many studies [70,71]. The most manifested phenotype induced by RNAi-*Djhsp60* is progressive head regression, but the underlying mechanism remains unknown. We demonstrated that RNAi-*Djhsp60* damages the structure and function of the mitochondria, and we also observed a large number of dead cells in RNAi animals, but which type of cell death occurred in the RNAi animals is a very interesting question. It is intriguing that we did not observe the upregulation of apoptosis-related genes, and we also did not observe that apoptotic cell death occurred in RNAi animals. However, we observed that necrotic-like cell death occurred in RNAi animals, along with the upregulation of cathepsin L transcripts. Cathepsin L, being a lysosomal cysteine proteinase, might also be secreted into the extracellular space. Previous studies revealed that cathepsin L is involved in the degradation of intracellular proteins and the breakdown of extracellular proteins, and programmed necrotic cell death is supposed to be related to cathepsin L activity [44,56,72]. Therefore, it is reasonable that mitochondrial dysfunction causes disruption of the lysosome, which releases cathepsin L and leads to necrotic-like cell death. As the intestinal branches can extend to the tip of the head and cathepsin L is mainly expressed in intestinal cells in planarians, we propose that head regression and head tissue degradation are related to cathepsin L activity (Figure 12B).

## 5. Conclusions

Our work demonstrated that HSP60 is necessary for planarian regeneration and homeostasis. On the basis of our results, we propose a model of HSP60 function in planarians (Figure 12). The knockdown of *Djhsp60* impairs the structure and function of the intestinal tissue, which cannot provide nutrients for neoblasts. Without nutrients from the intestinal cells, neoblasts cannot synthesize ATP and the proteins needed for proliferation, migration, and differentiation, further leading to the loss of regeneration abilities and homeostasis. In addition, mitochondrial dysfunction causes lysosome disruption in intestinal cells, releasing cathepsin L into the extracellular space, which eventually leads to head regression. Our work highlights the essential role of the intestinal system in planarian regeneration and homeostasis.

## Figures and Tables

**Figure 1 biomolecules-12-00808-f001:**
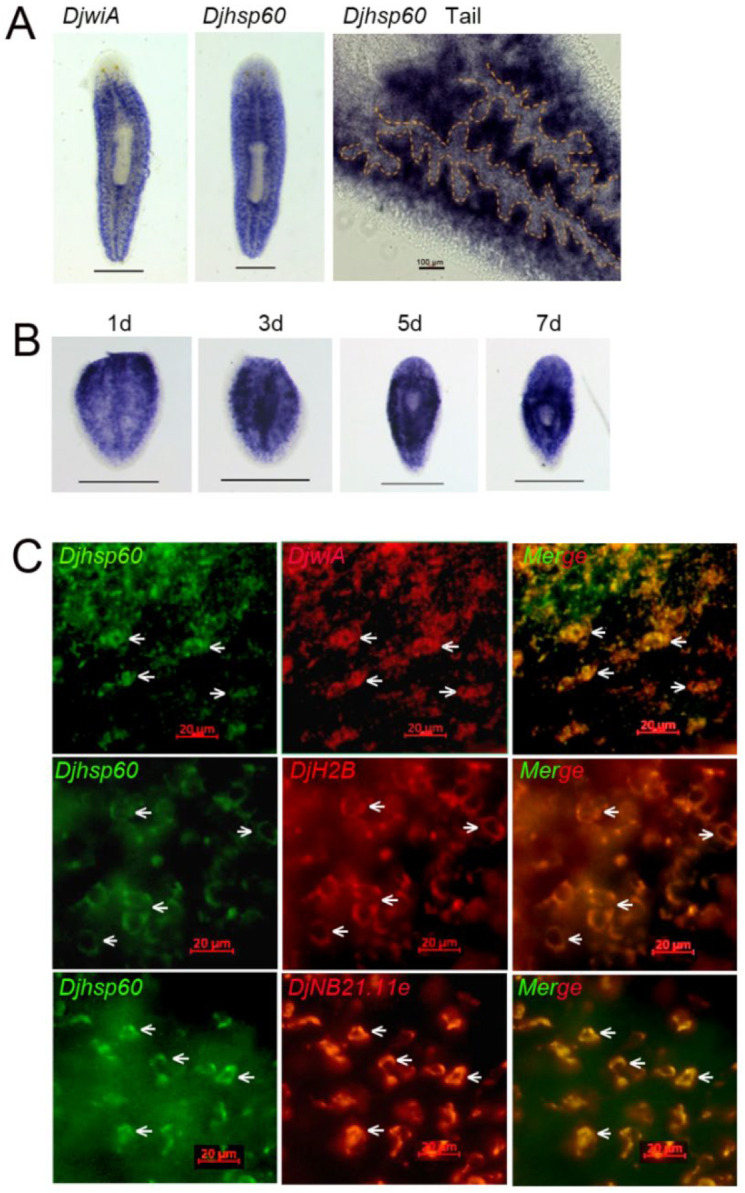
Expression patterns of *Djhsp60*. (**A**) Expression patterns of *Djhsp60* in intact animals revealed by WISH. The orange dashed line indicates the border between parenchymal tissues and intestinal tissue. *n* = 6 animals. Scale bar 0.5 mm. (**B**) Expression patterns of *Djhsp60* during planarian regeneration revealed by WISH. *n* = 6 animals. Scale bar 0.5 mm. (**C**) Co-expression of *Djhsp60*/*DjH2B* and *Djhsp60*/*DjNB21.11e* revealed by double-FISH. Arrows indicate the co-expressed cells. *n* = 5 animals. Scale bar: 20 μm.

**Figure 2 biomolecules-12-00808-f002:**
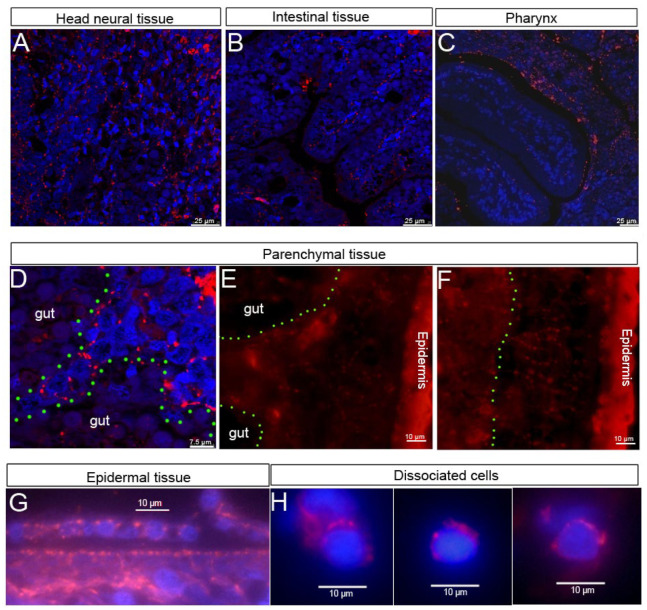
The distribution of DjHSP60 protein revealed by fluorescent immunostaining. (**A**–**D**) Horizontal sections captured by confocal microscope. (**E**–**G**) Sagittal sections captured by fluorescent microscope. (**H**) Dissociated cells captured by fluorescent microscope. DjHSP60 is located in cytoplasm. The green dashed line indicates the border between parenchymal tissues and intestinal tissues. Nuclear staining is shown in blue and HSP60 staining in red.

**Figure 3 biomolecules-12-00808-f003:**
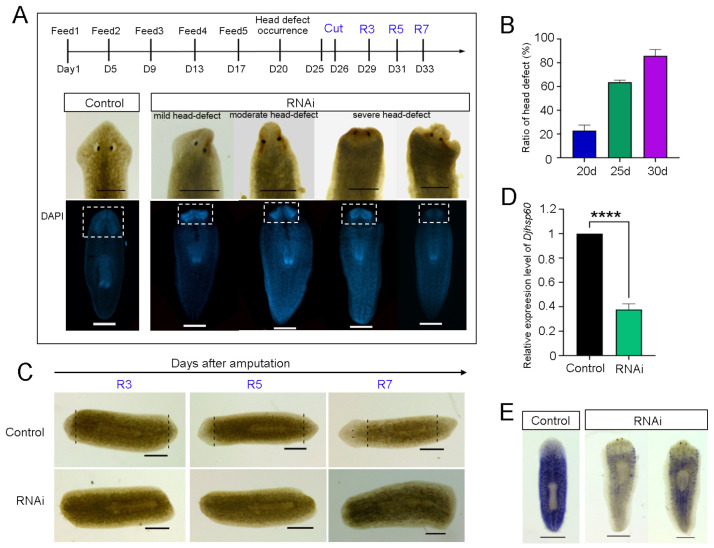
Phenotype observation of *Djhsp60*-RNAi animals. (**A**) Live images of head-defective animals and brain structure revealed by DAPI staining after 25 days of dsRNA exposure. Brain structure was boxed. (**B**) Ratio of head-defects occurrence after RNAi. Three individual RNAi experiments were used for statistics. (**C**) Effects of RNAi on planarian regeneration. (**D**) Endogenous mRNA expression levels of *Djhsp60* after RNAi was detected by qPCR. Asterisks indicate statistical differences (**** *p <* 0.0001). (**E**) Expression of *Djhsp60* after RNAi was shown by WISH. Color development for 45 min. *n* = 6 animals. Scale bar: 0.5 mm.

**Figure 4 biomolecules-12-00808-f004:**
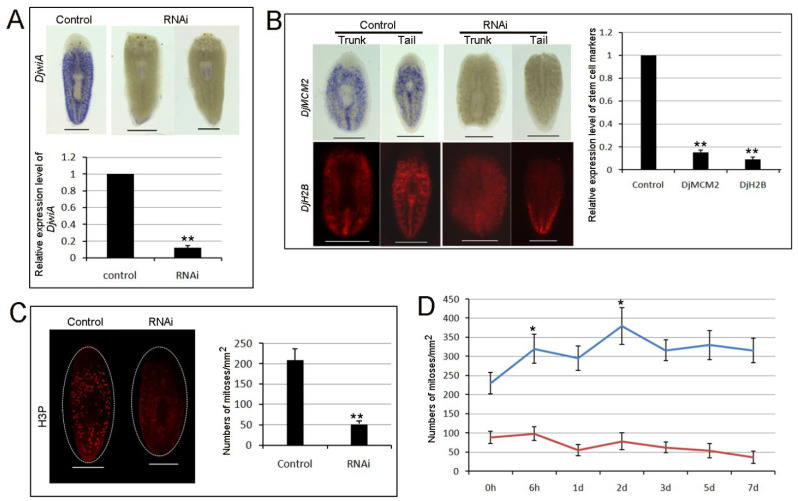
RNAi-*Djhsp60* affects stem cell function. (**A**) Expression of *DjwiA* after RNAi in intact animals revealed by WISH and qRT-PCR. Color development for 45 min. *n* = 6 animals. (**B**) Expression of stem cell markers revealed by WISH, FISH and qRT-PCR after RNAi at 5 days of regeneration. Color development for 45 min. *n* = 6 animals. (**C**) Representative of images labeled with anti-H3P and quantification of mitotic cells in intact control and RNAi animals. (**D**) Dynamic changes of mitotic cells after RNAi at different time points post-amputation. Asterisks indicate statistical differences (* *p* < 0.05, ** *p* < 0.01). Scale bar: 0.5 mm.

**Figure 5 biomolecules-12-00808-f005:**
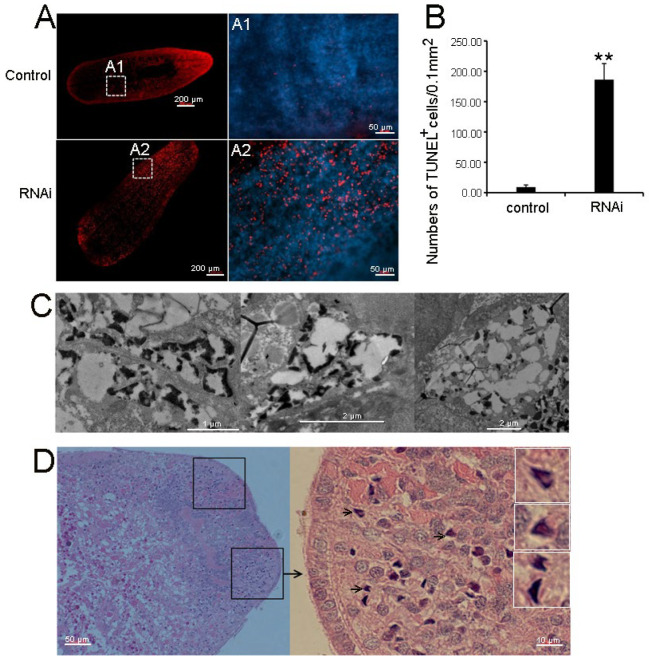
RNAi-*Djhsp60* induced cell death in planarians. (**A**) Representative images of TUNEL-stained intact animals. A1 and A2 indicate image magnification. (**B**) Quantification of TUNEL positive cells, *n* = 5 animals. Asterisks indicate statistical differences (** *p* < 0.01). (**C**) Necrotic-like cell death in RNAi animals revealed by TEM. (**D**) Necrotic-like cell death in RNAi animals revealed by HE staining. Black box indicates the degenerating tissue. Arrow indicates the dead cell. White boxes indicate the image magnification of the dead cells. RNAi-animals with mild-head defect were collected for HE staining and TEM.

**Figure 6 biomolecules-12-00808-f006:**
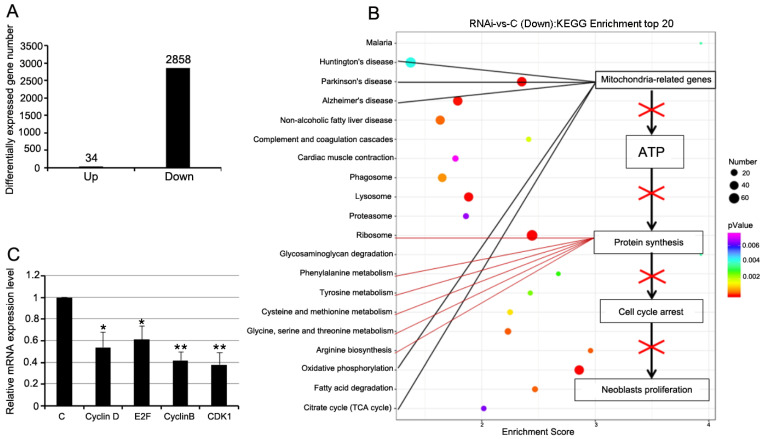
Differently expressed genes after RNAi revealed by RNA-seq. (**A**) Differently expressed genes number after RNAi. (**B**) KEGG enrichment of top downregulated genes. Based on this result, a hypothesis was proposed to explain RNAi phenotype. (**C**) Relative expression levels of cell cycle related genes detected by qRT-PCR. Asterisks indicate statistical differences (* *p* < 0.05, ** *p* < 0.01).

**Figure 7 biomolecules-12-00808-f007:**
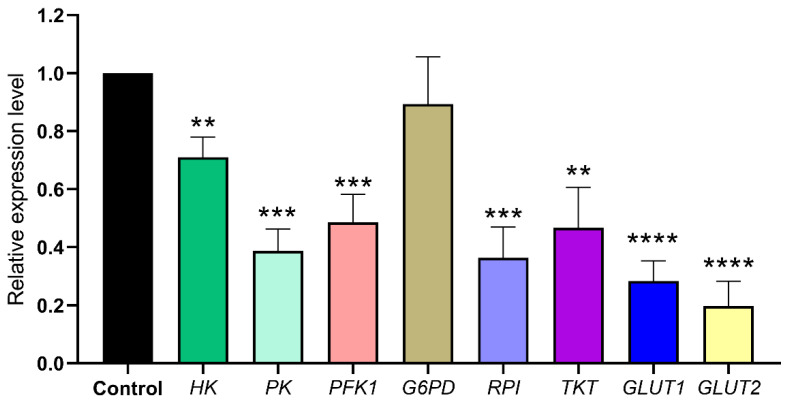
Relative expression levels of transcripts related to neoblasts energy metabolism detected by qRT-PCR. Asterisks indicate statistical differences (** *p <* 0.01, *** *p <* 0.001,**** *p <* 0.0001).

**Figure 8 biomolecules-12-00808-f008:**
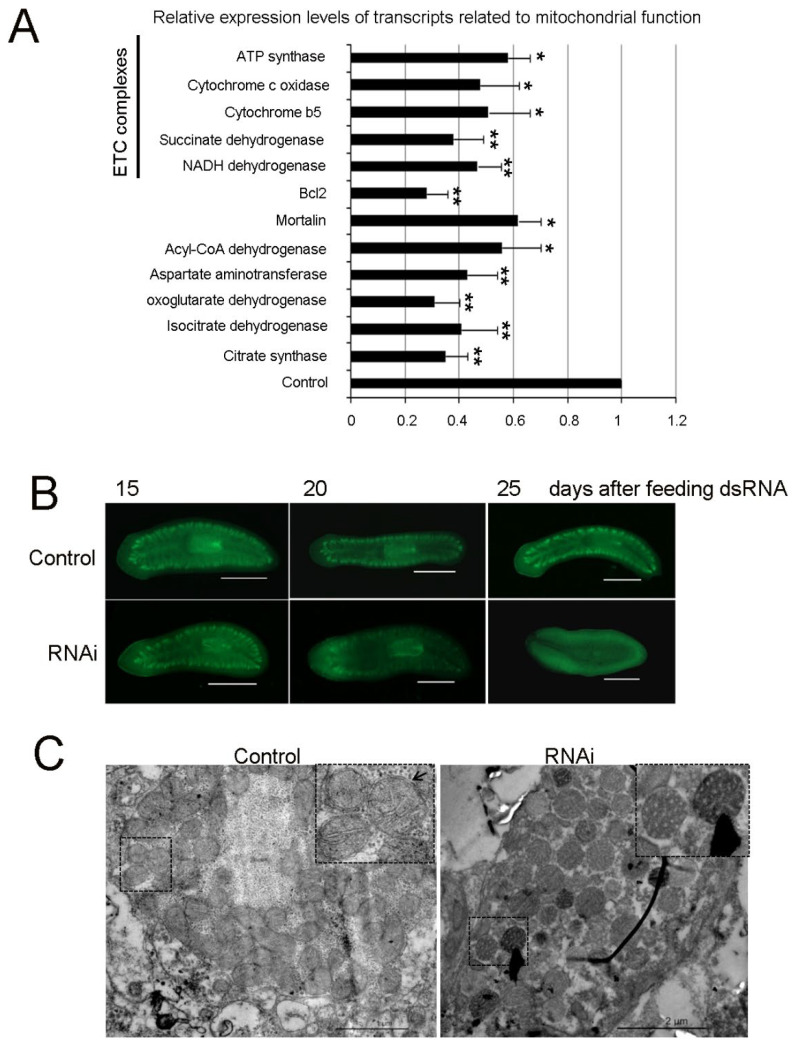
Mitochondrial function is severely impaired by RNAi-*Djhsp60*. (**A**) Relative expression levels of transcripts-related to mitochondrial function detected by qPCR. Asterisks indicate statistical differences (* *p* < 0.05, ** *p* < 0.01). (**B**) Assessment of mitochondrial respiration after the feeding of dsRNA revealed by H_2_DCFDA staining. Ventral views, *n* = 5 animals. Scale bar: 1 mm. (**C**) Ultrastructure of mitochondria revealed by TEM. Honeycomb-like mitochondria occurred in RNAi animals. Double membrane layers were indicated by black arrow.

**Figure 9 biomolecules-12-00808-f009:**
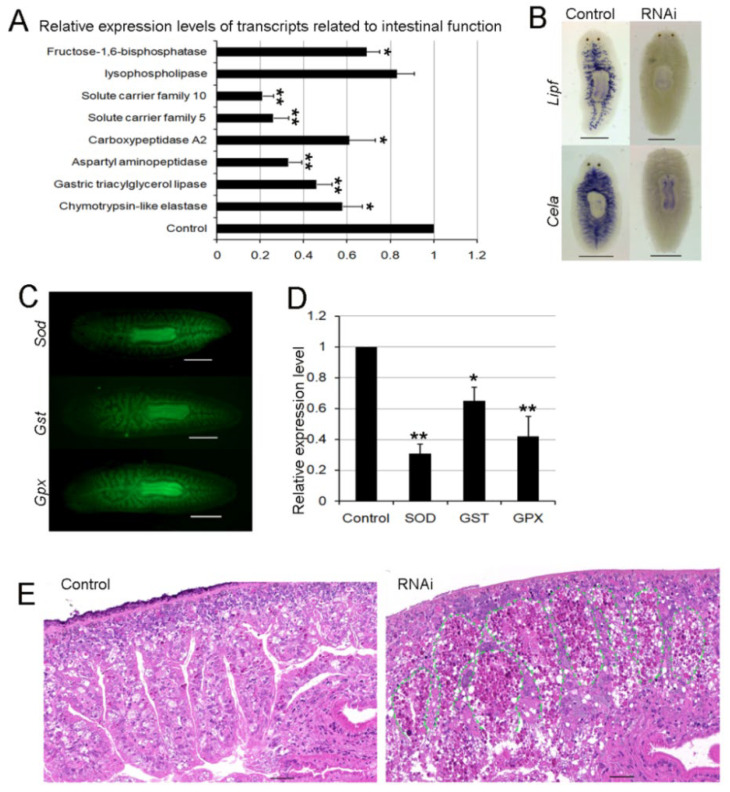
Effects of RNAi-*Djhsp60* on the structure and function of intestinal tissues. (**A**) Relative expression levels of transcripts related to intestinal function detected by qRT-PCR. Asterisks indicate statistical differences (* *p* < 0.05, ** *p* < 0.01). (**B**) Expression patterns of gastric triacylglycerol lipase (*Lipf*) and chymotrypsin-like elastase (*Cela*) after RNAi revealed by WISH. Color development for 30 min. Scale bar: 0.5 mm. (**C**) Expression patterns of antioxidant enzymes in normal animals revealed by FISH. Scale bar: 0.5 mm. (**D**) Relative expression levels of antioxidant enzymes after RNAi detected by qRT-PCR. Asterisks indicate statistical differences (* *p* < 0.05, ** *p* < 0.01). (**E**) Morphological changes of intestinal tissues after RNAi revealed by HE staining. The green dashed line indicates the border between parenchymal tissues and intestinal tissues. Scale bar: 50 μm.

**Figure 10 biomolecules-12-00808-f010:**
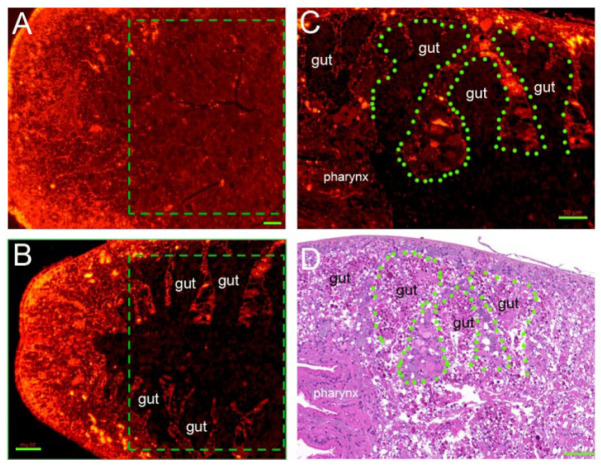
Distribution of HSP60 protein in control and RNAi-animals revealed by fluorescent immunostaining. Parts of differential region between control and RNAi were boxed. (**A**) Control section. (**B**–**D**) RNAi sections. (**C**,**D**) showed the same part of the body. The green dashed line indicates the border between parenchymal tissues and intestinal tissues. Animals with mild head-defects were collected for this experiment. Scale bar: 50 μm.

**Figure 11 biomolecules-12-00808-f011:**
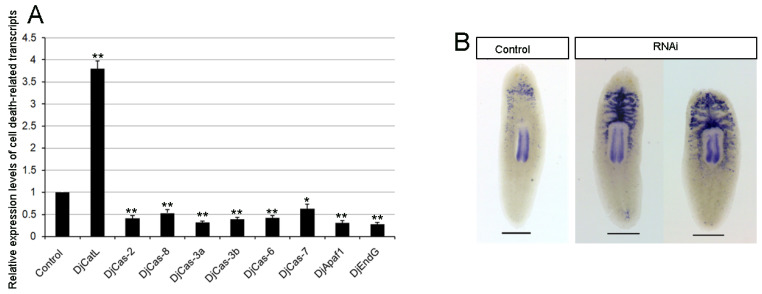
Effects of RNAi-*Djhsp60* on the expression of cell death-related transcripts. (**A**) Relative expression levels of cell death-related transcripts detected by qRT-PCR. Asterisks indicate statistical differences (* *p <* 0.05, ** *p <* 0.01). (**B**) Expression pattern of *DjCatL* after RNAi revealed by WISH. Color development for 45 min for this experiment. If color was developed for 2 h, CatL transcripts were distributed in intestinal system from head to tail, *n* = 5 animals. Scale bar: 0.5 mm.

**Figure 12 biomolecules-12-00808-f012:**
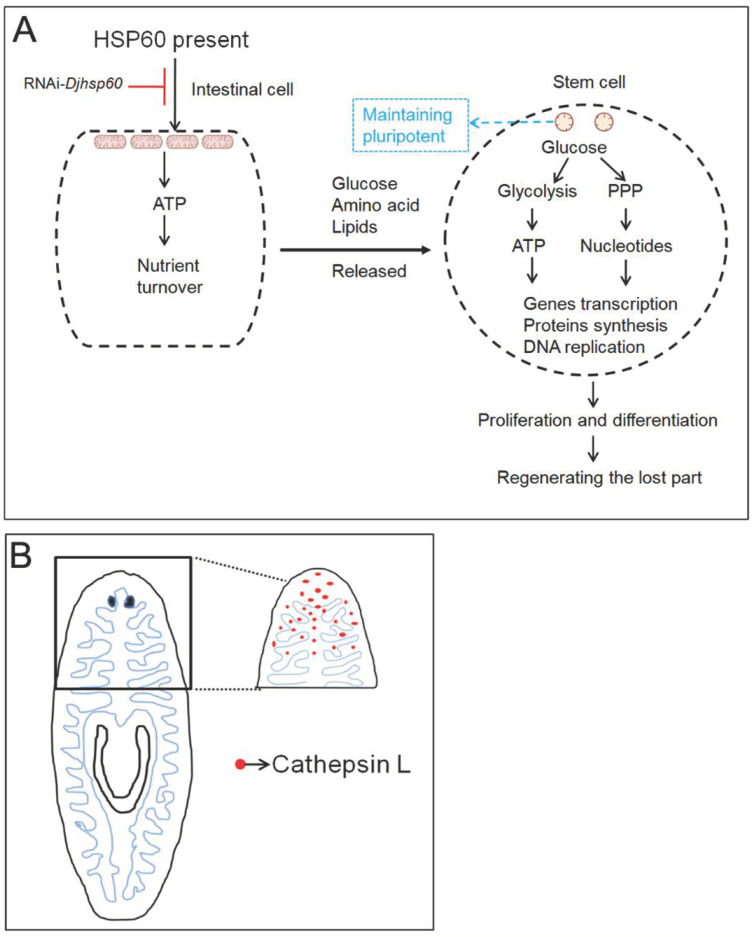
Proposed model illustrating the necessity of HSP60 in planarian regeneration and tissue homeostasis. (**A**) HSP60 functions in maintaining the integrity of mitochondria, which show distinct morphology and function in stem cells and intestinal cells. Intestinal cells depend on mitochondria to produce ATP for nutrient turnover, and to release nutrients for stem cells and other types of cells. Stem cells mainly depend on glycolysis to produce ATP for genes transcription and protein synthesis, and also depend on the pentose phosphate pathway (PPP) to produce nucleotides for DNA replication. Mitochondria in stem cells are related to maintaining pluripotency. (**B**) Intestinal branches can extend to the tip of planarian head. Activating and releasing of cathepsin L from intestinal cells induced by RNAi-*Djhsp60* may be the reason for head regression.

## Data Availability

All data are included in this manuscript or the Appendix A. Requests for data and materials should be addressed to Kexue Ma.

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
