# Peer review of "Djhsp60 Is Required for Planarian Regeneration and Homeostasis"

_biomolecules, 2022, doi:10.3390/biom12060808_

Round 1
Reviewer 1 Report
The manuscript by Ma and collaborators studies the function of the hsp60 chaperone in planarians. Silencing of the hsp60 chaperone perturbs planarian homeostasis and results in head regression and neoblasts depletion, as well as perturbs the regenerative process, and avoids blastema formation and stem cell proliferation. The authors perform RNAseq transcriptomic analysis on Hsp60 silenced animals. Interestingly, they found that only 34 transcripts appear upregulated on this transcriptomic analysis, while a general downregulation of the expression of genes related to mitochondrial, metabolic, intestinal and neoblast function is observed. One of those upregulated transcripts corresponds to the cathepsin L gene. Based on this data, the authors suggest that the effects of hsp60 silencing on planarian regeneration and homeostasis relies on the disruption of mitochondrial function and the activation of cathepsin-mediated cell death. From my point of view, the manuscript does not include the appropriate experiments to allow that conclusions about the mechanism, that remain largely speculative, and only correlations that need to be functionally validated are included. I consider that the manuscript needs further functional validation before it is ready for publication in the journal Biomolecules.
My major concerns are:
1/ The authors state that hsp60 mRNA is not only expressed in stem cells as previously reported but also in other differentiated tissues, although no expression of hsp60 mRNA in differentiated cells is shown in the manuscript. Expression of hsp60 mRNA with neoblasts and with an epidermal progenitor markers is shown. I wonder whether the authors also observed expression of hsp60 mRNA in other cellular (differentiated) types in this experiment. A further characterization of the expression pattern of hsp60 transcripts is needed to validate its expression on differentiated tissues. For instance, in situ hybridization on irradiated planarians could allow to conclude whether hsp60 mRNA is also expressed on planarian depleted of neoblasts, and thereby will permit to better observed whether this gene is expressed on differentiated tissues such as the parenchyma, the intestine or the nervous system. In addition, in situ hybridization on dissociated cells could validate expression of hsp60 mRNA on cells different than neoblasts. Finally, the analysis of the available single cell data from a close related planarian species could also be used to get a better idea on which cellular types express hsp60 mRNA in planarians.
2/ To further analyze hsp60 in planarians, the authors study the distribution of the HSP60 protein using a commercial antibody. The authors argue that this antibody stains ubiquitously in several differentiated tissues, an expected distribution for a protein that could be expressed in all planarian cell types. Using this antibody, in figure 2, they state that HSP60 protein is detected in head neural cells (2A), intestinal tissue (2B) and Pharynx (2C). These structures are really difficult to recognize in those images with a simple nuclear staining (I am unable to see the head neural tissue, for instance), and I wonder how the authors can conclude on such expression. Colocalization of HSP60 staining together with available antibodies against neural or intestinal planarian cell types will help the general reader to follow author´s statements. More importantly, further controls of the specificity of the antibody are needed. A control of the specificity of the antibody is shown at the end of the manuscript (Fig 9F). Although the authors argue that on hsp60 silenced treated planarians less cells stain with the HSP60 antibody, that conclusions are not clear to me. Additional and representative/detailed images on those differences would be appreciated. Indeed, I do not see a clear difference in the staining between controls and hsp60 RNAi silenced planarians and I observe a general reddish staining on both samples. A closer (cellular) view on those staining’s would be much appreciated. Have the authors essay whether HSP60 antibody detects planarian HSP60 by Western blot? Those experiments will further allow to analyze the specificity of the commercial HSP60 antibody and will allow to determine whether hsp60 RNAi results in a decay of HSP60 protein.
3/ In general, there is a lack of information regarding the time points for some of the experiments. For instance, an important information that is not included in the manuscript is the time point at which the authors perform the RNAseq experiment. As hsp60 silencing ends in lethality (again, the reader does not know at which time point the animals eventually die), this information is critical for the interpretation of the data. Information regarding the timing of experiments 3D, 3E, 4A, 4C, Figure 5, 6, 7, 8A, 8C, 9, 10 is also missing.
4/ As the authors argue, in planarians, head regression is known to be an indicator of neoblasts depletion (as happens after piwi RNAi , lethal irradiation… ) and it is caused by the perturbation of the normal homeostasis of the animal and the lack of cell turnover. So, head regression is associated with neoblasts depletion. Thereby, it is not surprising that neoblasts are depleted and that the regenerative capacity of the animal is perturbed in animals that suffer head regression (Day 26). Have the author assay if hsp60 silencing perturbs planarian regeneration at earlier time points? Is the neoblasts population affected at those early time points?
5/ At which time point TUNEL was assayed? And TEM? Again, that information is critical for the interpretation of the data. If planarians are dying, an increase in the number of TUNEL + cells is expected. Does necrotic cells are specific of hsp60 dying planarians? Would lethally irradiated planarians or piwi silenced planarians suffering head regression also show those necrotic cells? Also, the same region of the planarian should be compared in control and RNAi treated planarians (Figure 5A, the inset of controls and hsp60 silenced planarians does not show equivalent regions).
6/ Please, specify at which time point the RNAseq was performed, as downregulation of neoblasts, mitochondrial, metabolic and other genes might be expected in dying planarians. Does mitochondrial H2DCFDA assay detects perturbation of mitochondrial function/ presence of necrotic cells / expression of metabolic genes is affected before head regression (neoblast depletion) is observed? In addition, it is surprising that only 34 genes appear upregulated compared to 2858 downregulated. Which are the other 33 genes that appear upregulated? Those genes are completely ignored in the manuscript. Please, add a list of the upregulated genes on a supplementary table and include those genes on the discussion.
7/ The last sentence of the results section clearly overstate the results of the RNAseq data: ‘our results clearly demonstrate that cathepsin L is the death executer in RNAi-hsp60 animals’. The hypothesis that the overexpression of cathepsin L is related with the necrosis observed could be investigated by assaying, for instance, if silencing of cathepsin L rescues the necrotic phenotype of hsp60 silenced planarians. It would also be interesting to check whether not only caspase expression but caspase activation is altered in those conditions. On the other hand, it is interesting that cathepsin L expression is restricted to the anterior region of the digestive system in D japonica, but it is not restricted on the AP axis in other planarian species, such as S mediterranea. Could the authors discuss those differences in the expression pattern?
8/ Have the authors performed any positive (H2O2) or negative (ROS inhibitors) control of the mitochondrial respiratory assay? How do they arrive to the conclusion that hsp60 silencing mainly disrupts intestinal mitochondria? At which time points has the mitochondrial integrity and activity has been assayed? Before or after head regression is observed?
Minor comments:
-Figure 1: orange should be read instead of ¨organge¨
-Figure 2: please, specify that nuclear staining is shown in blue and HSP60 staining in red. Adding a schema or an image of the full section will help the reader to orient on those stainings.
-Figure 2 E and 2 F: there are two regions on the images that appear with different intensity/ color: I highlighted them on these captures:
-Figure 5, TEM and HE staining images from control animals should be added.
-Why only the genes that are downregulated in the RNAseq are shown in table Supl5? For instance, the Cathepsin L gene appears upregulated in the RNAseq of hsp60 and it is listed on Supl Table 5, but its fold change is not listed.
- Was the incubation of the antibody performed at 37 degrees as stated in section 2.5?
- The accession number of the genes used in the study would be appreciated. That information could be added in Supl table 1, for instance.
Reviewer 2 Report
Ma et al. have submitted a manuscript describing the role of hsp60 in Planarian regeneration and its impact on mitochondrial function and thereby cell death. RNAi of hsp60 leads to head regression and loss of regenerative abilities owing to decreased mitotic cells but upregulates expression of Cathepsin L-like gene leading to increased cell death. Overall the phenotype is well characterized, with multiple tools involving transcriptomics and Electron microscopy to reveal the effect on mitochondria. I have the following suggestions for the authors to consider in their revised version.
- In fig. 1C the authors show co-expression of djwiA and Djhsp60 but the images are very blurry to appreciate the staining. Please revise and put nuclear staining to identify individual cells.
- Can the authors put some controls (no primary and no secondary antibody controls) to confirm the specificity of the DJHSP60 antibody used for immunostaining in fig. 2. Similar concern for fig 9F (Put nuclear staining to understand cellular structure)
- Introduction-The authors discuss the proliferation and migration of neoblasts and the role of autophagy in stem cells. This sentence needs proper referencing- Guedelhoefer, Development, 2012 and Sahu et al., eLife 2021 discussing neoblast migration, Gonzalez Estevez and Salo 2009 review in Autophagy.
- In fig. 3C, the authors show the regeneration defect in RNAi animals from the trunk. Does tail regeneration also affect the head fragments?
- RNAseq is performed on bulk animals or on FACS sorted stem cells? Please mention the number of animals used and replicates for RNAseq.
- The TEM images show mitochondrial ultrastructure, which cell are we looking at-stem cells or intestinal cells, or other cell types? Also mention how many micrographs were studied to confirm mitochondria are affected in RNAi animals.
- DjCatL is highly upregulated in the intestinal branches near the head region and authors claim the increased expression leads to head regression. I am little concerned with this claim as intestinal branches also lead to the tail and thereby should also lead to tail lysis? One way to tackle this is to perform beta-catenin RNAi (leading to double head) and APC RNAi (leading to double tail) and check for head regression and Cathepsin L expression.
- Typo in figure legend 8C – “Ultrastructure of mitochondria”.
I apologize to the authors that with MDPI formatting I am unable to access their supplementary figures.
Author Response
The authors have extensively tried to improve the manuscript and address my concerns with the current tools available. I recommend its publication.
Answer:
Dear reviewer,
We are very grateful to you for your critical comments on our manuscripts. Your instruction and suggestion are really appreciated, and help us improve the quality of our manuscript.
In addition, we invite MDPI English Editing group to revise English language. Please see the revised manuscript.
Thank you again!
Best regards
Kexue Ma

Reviewer 3 Report
in this manuscript, authors investigate the role of DjHSP60 in head regeneration of planarians. they obsreved that hsp60 RNAi induce head regression. the knock down of RNAi-hsp60 disrupts the structure of mitochondria, inhibits the mito-chondrial-related genes, and damages the integrity of intestinal tissues via a down-regulation the intestinal-expressed genes. Authors suggest a model illustrating the rela-tionship between neoblasts and intestinal cells, unravelling the essential role of intestinal system in planarian regeneration and tissue homeostasis.
the manuscript is well done, experiments and conclusion are well presented and conclusive.
Minor comments
figure 1C, midel panel (DJH2B experiemnts) looks blur
Round 2
Reviewer 1 Report
The additional information provided by the authors to characterize the specificity of the antibody is appreciated. However, I still have some concerns about it. The authors show that HSP60 protein disappears from intestinal tissues after Hsp60 RNAi (Figure 10B-C). Based on those images, in hsp60 silenced planarians, HSP60 exclusively disappears from the intestinal tissue, as strong red signal similar to the one observed in control planarians is visible in the head region of hsp60 silenced planarians. How the authors explain that the RNAi exclusively affects the planarian intestinal tissue and does not perturb HSP60 protein in other tissues after 25 days of treatment? Has this tissue specificity been previously observed after silencing other genes in this model organism?
From my point of view, overall, this manuscript shows that hsp60 is needed for planarian survival, and no effects of hsp60 silencing are observed or characterized previous to the regression of the planarian head and the depletion of the neoblasts. Most effects observed by the authors after hsp60 silencing such as perturbation of regeneration, depletion of mitochondrial activity, increased cell death assayed by tunnel assay, appearance of necrotic cells, cathepsin overexpression, … exclusively appear in planarians that have started to die due to neoblasts depletion after 25 days of treatment. Although it seems clear that Hsp60 silencing perturbs the planarian gut, I wonder whether the authors have checked if intestinal genes appear altered at any early time point before head regression starts. If nutrient deprivation and intestine integrity are needed for the survival of planarian neoblasts, we expect perturbation of intestinal tissue previous to head regression and neoblasts depletion. It seems to me that the time point chosen for the experiments (25 days, so planarians that have already started to die) allow the characterization of the latest’s events leading to planarian death due to neoblasts depletion instead of a more specific effect of hsp60 silencing. As the authors also point in their response, those effects do not seem different to the ones caused by neoblasts depletion by X-ray or piwi silencing. Finally, I do consider that this manuscript does not provide the information necessary to concluded that the death of hsp60 silenced planarians is mediated by cathepsin (especially if the authors checked that possibility and could discard it experimentally) and such conclusion should be removed from the title.
Lastly, I would like to point out that comments made by reviewers are made to improve the manuscript. In order to do so, I consider that all the additional information provided to answer the referee’s concerns should be added to the manuscript and not exclusively used in the response to the reviewers. For instance, in addition to the further characterization of the expression pattern of Hsp60 with the analysis of available SC data, the authors should add to the manuscript the information regarding the assays to check for the specificity of the HSP60 antibody, the results of the ´no´effect of hsp60 silencing on planarian regeneration at early time points (before neoblasts depletion by homeostasis leading to head regression), the impossibility to rescue the hsp60 silencing phenotype by cathepsin RNAi ….
Other comments:
Figure 4B, H2B staining is blurry,
Figure 5A, TUNEL images are not of enough quality. Is there any reason for the strongest reddish staining in the posterior region of the animals? Why do the DAPI staining is barely visible in figure A1 (control) compared to A2 (hsp60 RNAi)? How were TUNEL positive ´cells´ could be quantified if the nuclei are not visible on those images?
Reviewer 2 Report
The authors have extensively tried to improve the manuscript and address my concerns with the current tools available. I recommend its publication.
